# Rational Design of Novel Single-Atom Catalysts of Transition-Metal-Doped 2D AlN Monolayer as Highly Effective Electrocatalysts for Nitrogen Reduction Reaction

**DOI:** 10.3390/molecules29235768

**Published:** 2024-12-06

**Authors:** Xiaopeng Shen, Qinfang Zhang

**Affiliations:** 1School of Chemistry & Chemical Engineering, Yancheng Institute of Technology, Yancheng 224051, China; 2School of Materials Science and Engineering, Yancheng Institute of Technology, Yancheng 224051, China

**Keywords:** electrochemistry, nitrogen reduction reaction (NRR), single-atom catalysts, 2D AlN monolayer

## Abstract

The single-atom catalysts (SACs) for the electrocatalytic nitrogen reduction reaction (NRR) have garnered significant attention in recent years. The NRR is regarded as a milder and greener approach to ammonia synthesis. The pursuit of highly efficient and selective electrocatalysts for the NRR continues to garner substantial interest, yet it poses a significant challenge. In this study, we employed density functional theory calculations to investigate the stability and catalytic activity of 29 transition metal atoms loaded on the two-dimensional (2D) AlN monolayer with Al monovacancy (TM@AlN) for the conversion of N_2_ to NH_3_. After screening the activity and selectivity of NRR, it was found that Os@AlN exhibited the highest activity for NRR with a very low limiting potential of −0.46 V along the distal pathway. The analysis of the related electronic structure, Bader charge, electron localization function, and PDOS revealed the origin of NRR activity from the perspective of energy and electronic properties. The high activity and selectivity towards the NRR of SACs are closely associated with the Os-3N coordination. Our findings have expanded the scope of designing innovative high-efficiency SACs for NRR.

## 1. Introduction

Ammonia (NH_3_) not only plays an indispensable role in agriculture and chemical industry for human society [1,2,3] but also serves as a carbon-free hydrogen carrier due to its high hydrogen mass fraction (17.6 wt%) [4]. In the natural nitrogen cycle, the production of NH_3_ can occur through biological fixation of N_2_ by nitrogenase enzymes in bacteria, or through lightning-induced nitrogen fixation in the atmosphere. While in the traditional industrial synthetic NH_3_ industry, the Haber–Bosch (H-B) process is normally catalyzed within the Fe- or Ru-based catalysts under high temperature and pressure (700 K and 100 bar) react conditions [5], it consumes a massive quantity of fossil fuels and releases substantial amounts of carbon dioxide, thus causing energy crisis and serious environmental problems [6]. So, developing an environmentally friendly and cost-effective alternative to the Haber–Bosch process for NH_3_ production is a pressing task.

In recent years, the electrocatalytic nitrogen reduction reaction (eNRR) under ambient conditions has aroused the extensive interest of researchers [7,8,9,10,11,12,13,14]. The stable and efficient catalysts are crucial for eNRR as they help in reducing the energy barrier of the reaction. Among the diverse range of catalysts, single-atom catalysts (SACs) are emerging as the most promising. SACs process excellent catalytic stability, selectivity, and activity due to the strong electronic interaction between the single atom and the substrate. Currently, SACs have been widely applied in numerous important electrochemical catalytic reactions such as the NRR [10,11,12,13,14], the CO_2_ reduction reaction (CO_2_RR) [15,16,17,18], the oxygen reduction reaction (ORR) [19,20,21], the hydrogen evolution reaction (HER) [22,23,24,25], and the oxygen evolution reaction (OER) [26,27,28,29].

Many SACs composed of two-dimensional (2D) materials serve as the substrate for anchoring transition metal (TM) atoms and have been extensively studied for eNRR, such as TM@BN [30,31], TM@h-C_3_N_4_ [32], TM/g-C_9_N_10_ [12], TM-graphyne [33], TM@B_36_ [34], TM@MoSe_2_ [11], TM@BP [35,36], etc. These TM-SACs studies have proved that the empty d orbitals of TM atoms would accept the lone pair electrons of N_2_, while the electrons from TM-atom-occupied d-orbitals would be back-donated to the empty antibonding orbitals of N_2_, forming the “σ-donation and π-back-donation” electron transfer mechanism. However, the catalytic performance of the above-reported SACs is significantly inferior to that of traditional thermal ammonia synthesis catalysts, and it is still a long way from practical large-scale application. Moreover, metal atoms tend to migrate and aggregate to form metal nanoclusters while doped on the surface of the support, which would result in catalyst deactivation. Therefore, it is crucial to identify appropriate substrates that can provide TM-N coordinate sites to accommodate single TM atoms, improving the NRR performance of SACs.

The aluminum nitride (AlN) monolayer has been widely investigated, and it can be applied to high-power, high-temperature, and high-frequency electronic devices due to its broadband gap and high chemical and thermal stability [37]. To date, Zhang et al. have searched more than a dozen transition metal (TM)-doped AlN systems for HER/OER/ORR electrocatalysts based on first-principles calculations. They found that Co@AlN-VAl can be available as OER/ORR bifunctional catalysts with an η_OER_/η_ORR_ of 0.5/0.33 V, and Pd@AlN-VAl shows the highest HER catalytic activity with an η_HER_ of 0.009 V [38]. Wang et al. theoretically designed a bilayer tetragonal AlN monolayer as the Li-O_2_ electrochemical catalyst [39]. AlN monolayer and SiC emerge as promising catalysts for formic acid (HCOOH) production in CO_2_RR [40]. The above-reported studies have demonstrated that AlN nanomaterials have great potential value in the field of electrocatalysis, but there has been no relevant literature about AlN catalysts towards NRR until now.

Here, in this work, we introduce 29 kinds of TM atoms into the AlN monolayer with Al vacancy and systematically investigate the stability, catalytic activity, pathways, and selectivity of NRR using density functional theory (DFT) calculation. The Os@AlN SAC are screened out, and the theoretical limiting potentials are calculated to be −0.46 V by a distal mechanism. We further analyzed the electronic properties, electron transfer, electron localization function, and PDOS to explore the origin of the exceptional NRR performances. This work will provide theoretical guidance for designing advanced AlN-based materials for NH_3_ synthesis.

## 2. Computational Details

In this paper, the Vienna Ab initio Simulation Package (VASP) [41,42] is utilized to carry out all spin-polarized energy and structure optimization calculations. The Projector Augmented Wave (PAW) method is employed to delineate the electron–ion interactions [43,44]. The interaction between ions and electrons is illustrated using the Perdew–Burke–Ernzerh (PBE) functional within the generalized gradient approximation (GGA) [42]. The long-range van der Waals interaction between the adsorbates and the substrates is evaluated by the DFT-D3 method [45]. A vacuum space of 15 Å along the z-direction is added to sufficiently avoid interactions between periodic layers. For the structural optimization, the 4 × 4 supercell of AlN monolayer containing 32 atoms is constructed, in which the kinetic cutoff energy of the plane wave function is set as 450 eV, the convergence threshold of the energy and force are 10^−5^ eV and 0.02 eV/Å, respectively. The Brillouin zone is sampled by 3 × 3 × 1 and denser 21 × 21 × 1 Monkhorst–Pack (MP) grid during structural optimization and electronic properties calculations.

The binding energy (*E*_b_) can be calculated using the following equation:*E*_b_ = *E*_TM@AlN_ − *E*_AlN_ − *E*_TM_(1)
where *E*_TM@AlN_ is the total energy of the TM supported on the AlN monolayer, and *E*_AlN_ and *E*_TM_ are the energies of the AlN monolayer and a single TM atom, respectively.

The adsorption energy (*E*_ads_) of reaction intermediates on the TM@AlN catalysts can be calculated as follows:*E*_ads_ = *E*_total_ − *E*_TM@AlN_ − *E*_adsorbate_(2)
where *E*_total_ is the total energy of the adsorbate and TM@AlN species, and *E*_TM@AlN_ and *E*_adsorbate_ are the energies of TM@AlN and the free adsorbate, respectively. All energies were calculated with the same parameter settings. By definition, a negative *E*_b_ and *E*_ads_ value means that the adsorption is exothermic.

The changes in Gibbs free energy (Δ*G*) of each hydrogenation step in the NRR process were calculated by using the computational hydrogen electrode (CHE) model developed by Nørskov and co-workers [46], and Δ*G* could be determined as follows:Δ*G* = Δ*E* + Δ*E*_ZPE_ − *T*Δ*S* + Δ*G*_U_ + Δ*G*_pH_(3)
where ∆*E* is the reaction energy between the difference adsorption states of the intermediates, and Δ*E*_ZPE_ and ∆*S* are the changes in zero-point energies and entropy during the reaction, respectively. T is the temperature of 298.15 K. Δ*G*_U_ is the contribution of the applied electrode potential (U) to Δ*G*, respectively. Δ*G*_pH_ is the correction of free energy for H^+^, which was described by Δ*G*_pH_ = ln10 × *k*_B_*T* × pH, and *k*_B_ is the Boltzmann constant under standard reaction conditions (pH = 0, 298.15 K, 101,325 Pa). The Δ*G* values can be obtained from VASPKIT (https://vaspkit.com) [47]. The entropies of the gas molecules (i.e., H_2_, N_2_, and NH_3_) were taken from the NIST database.

The limiting potential (*U*_L_) was defined as *U*_L_ = −Δ*G*_max_*/e*, where Δ*G*_max_ is the maximum positive value of Δ*G*. *U*_L_ was the smallest applied negative potential to keep every elemental step becomes spontaneous. Therefore, it was utilized to assess the intrinsic NRR activity of catalysts. Ab initio molecular dynamics (AIMD) simulations were performed to evaluate the thermodynamic stability of the catalyst.

## 3. Results and Discussion

### 3.1. Screening of TM@AlN SACs as NRR Electrocatalysts

Firstly, we calculated the structures of the 4 × 4 × 1 supercell of the AlN monolayer anchoring the N_2_ molecule with end-on and side-on configurations. As illustrated in Figure 1 and Appendix A, the two structures have small negative adsorption energies (−0.093 eV and −0.083 eV) and large Al-N distances (2.727 Å, 3.586 and 3.537 Å between the AlN and N_2_ molecule, suggesting that the pristine 2D AlN monolayer is not suitable for electrocatalysis NRR. Therefore, we select the one Al-atom-defective AlN monolayer to anchor a single 3d, 4d, and 5d TM atom to construct the SACs candidates for catalyzing the NRR.

Then, after fully optimized without any constraints, the calculated binding energies of these 29 different SACs are listed in Figure 1b–d. The nonmagnetic (NM) and the ferromagnetic (FM) spin polarization have been involved in our calculation of the system energy. Our calculations show that the magnetic moments resulting from spin polarization mainly localize on the TM (Appendix A). The results show that most of the TM@AlN SACs are quite thermodynamically stable, while the Y@AlN is poorly stable, so this system is unsuitable as an electrocatalyst and will not be studied subsequently. It is worth mentioning that these large negative binding energies (ranging from −4.89 to −15.91 eV) are mainly due to the sp^2^ orbital hybridization of the N atom. Previous studies have shown that the entire NRR process comprises six intricate hydrogenation reaction steps with two types of N_2_ molecule adsorption models (end-on and side-on patterns) [1,48,49], and the free energy change in the first protonation (Δ*G*(*N_2_–*NNH)) or last protonation steps (Δ*G*(*NH_2_–*NH_3_)) exhibited the highest Δ*G* values among the various NRR pathways. We could use the screening criteria that both the values of the free energy change in the first protonation (Δ*G*(*N_2_–*NNH)) or last protonation steps (Δ*G*(*NH_2_–*NH_3_)) should be less than 0.49 eV for the extensive screening of highly active NRR catalysts [49]. The calculated Δ*G*(*N_2_–*NNH) and Δ*G*(*NH_2_–*NH_3_) of TM@AlN systems are shown in Figure 2. The results indicate that only the Os@AlN catalyst fulfilled the necessary criteria, and thus, it was subjected to further investigation and discussion.

### 3.2. NRR Mechanisms and Selectivity on Os@AlN

According to recently reported references [50,51,52,53,54], there exist four common NRR mechanisms, namely the distal, alternating, consecutive, and enzymatic reaction pathways. The distal and alternating pathways originate from N_2_ molecule end-on adsorption patterns, whereas the pathways of consecutive and enzymatic commence from N_2_ molecule side-on adsorption patterns (Appendix A). The optimized structures of the intermediates for each elementary reaction step in the four NRR reaction pathways are presented in Figure 3 and Figure 4. The Gibbs free energy diagrams of NRR processes on Os@AlN are presented in Figure 5. The limiting potential (*U*_L_) represents the minimum energy needed for NRR, which is determined by the potential determining step (PDS) with the highest Δ*G* among all reaction steps. The smaller the *U*_L_, the smaller the overpotential and energy barrier. Therefore, the corresponding reaction pathway is deemed as the optimal one, favoring the NRR process.

As displayed in Figure 5a,b, the last hydrogenation reaction can be viewed as the PDS of the distal pathway of NRR on Os@AlN. For the alternating pathway, the PDS is the second hydrogenation reaction. And the corresponding *U*_L_ are −0.46 V and −0.84 V, respectively. For the consecutive pathway of the Os@AlN catalyst, the most positive Gibbs free energy change (Δ*G*_max_) in the *N-*NH_3_→*NH reaction step with *U*_L_ of −0.77 V (Figure 5c). Along the enzymatic pathway, the PDS is the third hydrogenation reaction step with *U*_L_ = −0.86 V (Figure 5d). In brief, the Os@AlN is predicted to be a high-activity NRR electrocatalyst with a lower *U*_L_ value of −0.46 V through the distal pathway.

The selectivity of the Os@AlN catalyst is also evaluated by considering the competitive reaction of HER. An ideal catalyst for NRR is one that can effectively inhibit the HER. As depicted in Appendix A, the value of 
$\Delta G_{*N_{2}}$ (end on) (−0.074 eV) is more negative than the value of 
$\Delta G_{*H}$ (−0.042 eV), which infers that the N_2_ molecule is preferentially adsorbed on the catalyst than the H atom. This implies that the Os@AlN catalyst exhibits excellent NRR selectivity.

### 3.3. Origin of High NRR Catalytic Activity of Os@AlN

To explore the mechanism of catalytic activity from the perspective of electronic properties of Os@AlN, calculations of the density of states (DOS) and Bader charge analysis were performed. When compared with the DOS of free N_2_ (Figure 6a), the N_2_ molecule adsorbs on the Os@AlN catalyst with end-on adsorption pattern. This causes the antibonding state of N_2_ to split and hybridize with the Os-d orbital, resulting in occupied orbitals below the Fermi level (E_f_) and unoccupied antibonding states above E_f_. Figure 6c,d displays the PDOS before and after N_2_ adsorption on Os@AlN, with the Os-d and N-p orbitals marked in purple and blue, respectively. It can be clearly seen that the N-p orbital and the Os-d orbital are remarkably overlapped around the E_f_. This is attributed to the “acceptance–donation” mechanism [55], which is that the empty d orbitals of TM receive the lone-pair electrons from N_2_ and simultaneously donate the d orbital electrons back to the antibonding orbitals of N_2_. As a result, the inert N≡N bond is activated, making subsequent hydrogenation steps easier.

Furthermore, Appendix A (Appendix A) illustrates the band structures of pristine AlN, defective AlN with Al monovacancy, Os@AlN, and N_2_ adsorption on Os@AlN with end-on configuration. The pristine AlN is a semiconductor with an indirect band gap of 2.93 eV (Appendix A, Appendix A), which is smaller than the experimental value of 6.2 eV [56]. The outcome is widely recognized that DFT calculations tend to underestimate the semiconductor band gaps. While creating the monovacancy AlN, the original large band gap reduces to a small one (0.44 eV) (Appendix A, Appendix A). After the N_2_ adsorption on Os@AlN with end-on configuration, the band gap decreases from 1.22 eV (Appendix A, Appendix A) to 0.23 eV (Appendix A, Appendix A). This is due to the new Os-N bond being formed between the N_2_ molecule and the Os@AlN. To gain a deeper understanding of the interaction between the single Os atom and its neighboring N atoms, electron localization function (ELF) analysis was performed for Os@AlN and N_2_ adsorbed on Os@AlN with end-on configuration systems. In general, a high ELF value denotes a covalent bond type between atoms, while a lower ELF value indicates an ionic bond. It can be clearly seen from Figure 7a that when Os is anchored in the Al vacancy of the AlN monolayer, there is weaker electron localization between Os and its surrounding three N atoms, indicating the formation of the Os-N ionic bonds. After adsorbing the N_2_ molecule, a stronger electron localization between Os and its neighboring N atom can be found in Figure 7b, suggesting that fewer electrons are transferred from the N atom to Os and more electrons are transferred from the N_2_ molecule to Os.

Figure 8a illustrates the charge transfer of each component of Os@AlN SAC along the distal pathway, where the SAC system is divided into three moieties: the AlN substrate (moiety 1), the catalytic active center of Os atom and its three neighboring N atoms (moiety 2), and the N_x_H_y_ intermediates (moiety 3). Noticeably, a large number of electrons are transferred from the AlN substrate (moiety 1) to the N_x_H_y_ intermediates (moiety 3) through the Os-N3 active site (moiety 2). This indicates the AlN substrate serves as a significant source of electrons, while the Os-N3 active site can be regarded as the electronic transmitter. Figure 8b shows the differential charge density of N_2_ adsorbed on Os@AlN with end-on adsorption configuration. It reveals the accumulation and depletion of electrons at both the reactive site and the N_2_ molecule, indicating a bidirectional electron transfer between the TM atom and N_2_. The Bader charge analysis reveals that 0.41 e^−^ are transferred from Os@AlN to N_2_, effectively activating N_2_ and facilitating the subsequent hydrogenation process. At last, we conducted the ab initio molecular dynamics (AIMD) simulation at 350 K for 10 ps with a time step of 2 fs using a Nosé–Hoover thermostat to check the thermal stability of the Os@AlN system. Appendix A presents the variations in temperature and energy with time for Os@AlN. It has been observed that the temperature fluctuates slightly around 350 K, and the energy varies within a narrow range. Appendix A demonstrates that the geometric structure of the Os@AlN catalyst is well-preserved, with no evident structural distortion or collapse, thereby indicating its reliable thermodynamic performance.

## 4. Conclusions

In summary, we systematically investigated the SACs model of a single transition metal atom loaded on the AlN monolayer (TM@AlN) as the potential NRR electrocatalyst using DFT calculations. After calculating the N_2_ adsorption energies and activation and evaluating the reaction mechanism of the NRR process and selectivity between HER and NRR, Os@AlN is selected as a highly active NRR catalyst with a limiting potential of −0.46 V through the distal mechanism. The density of states, charge transfer, differential charge density, and electron localization function are utilized to uncover the source of NRR activity in Os@AlN. The AIMD simulation demonstrates that Os@AlN is thermodynamically stable. Our results indicate that AlN-based nanomaterials hold significant promise as an NRR electrocatalyst. This work offers novel insights for the rational design of effective SACs for NRR.

## Figures and Tables

**Figure 1 molecules-29-05768-f001:**
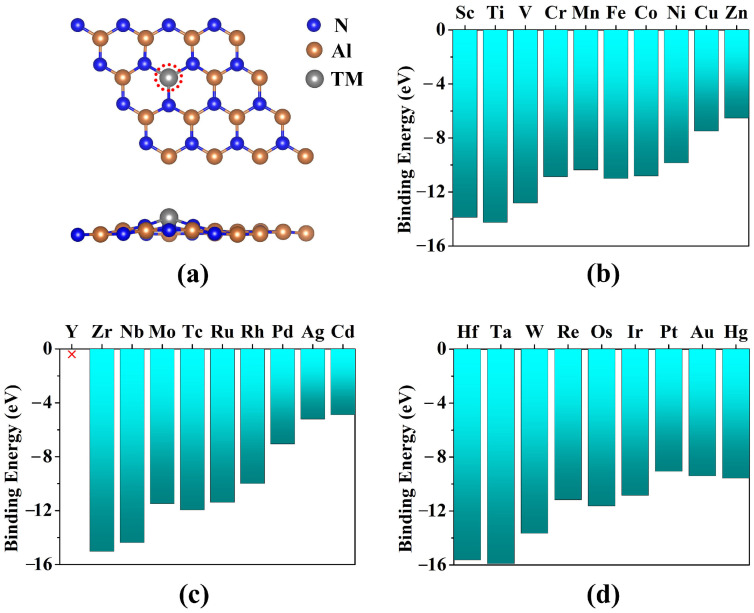
(**a**) The geometric structures of TM atoms anchored on the AlN monolayer with Al monovacancy. The anchoring sites are encircled in red colors. The binding energies of (**b**) 3d, (**c**) 4d, and (**d**) 5d TM atoms doped the AlN monolayer, respectively.

**Figure 2 molecules-29-05768-f002:**
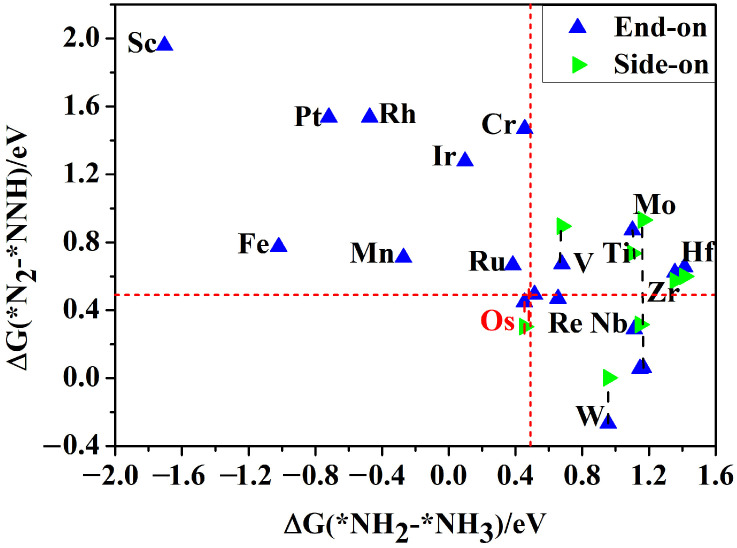
Screening results of TM@AlN for the NRR based on the free energy changes in the first hydrogenation step (△*G*(*N_2_–*NNH) with end-on or side-on configurations and last hydrogenation steps (△*G*(*NH_2_–*NH_3_)), respectively.

**Figure 3 molecules-29-05768-f003:**
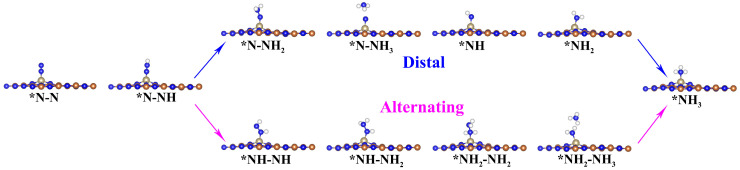
The optimized geometry structures of key intermediates for NRR on the Os@AlN with N_2_ end-on adsorption configuration.

**Figure 4 molecules-29-05768-f004:**
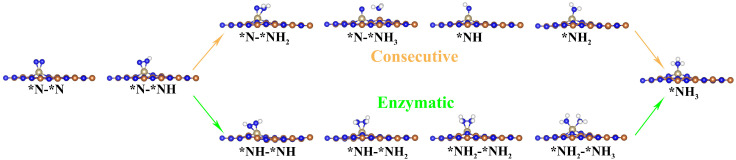
The optimized geometry structures of key intermediates for NRR on the Os@AlN with N_2_ side-on adsorption configuration.

**Figure 5 molecules-29-05768-f005:**
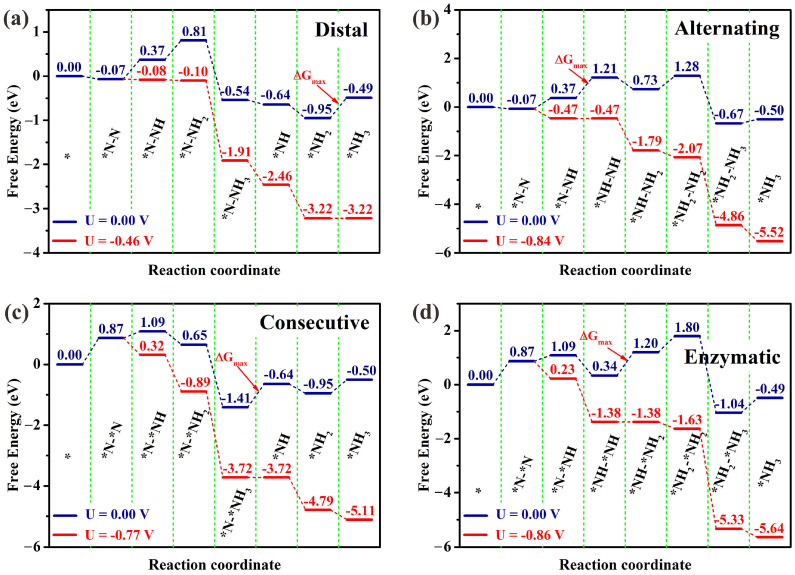
Gibbs free energy diagrams of NRR processes at zero (blue lines) and applied potential (red lines) via the (**a**) distal, (**b**) alternating, (**c**) consecutive, and (**d**) enzymatic pathways on Os@AlN, respectively. * represents the origin Os@AlN.

**Figure 6 molecules-29-05768-f006:**
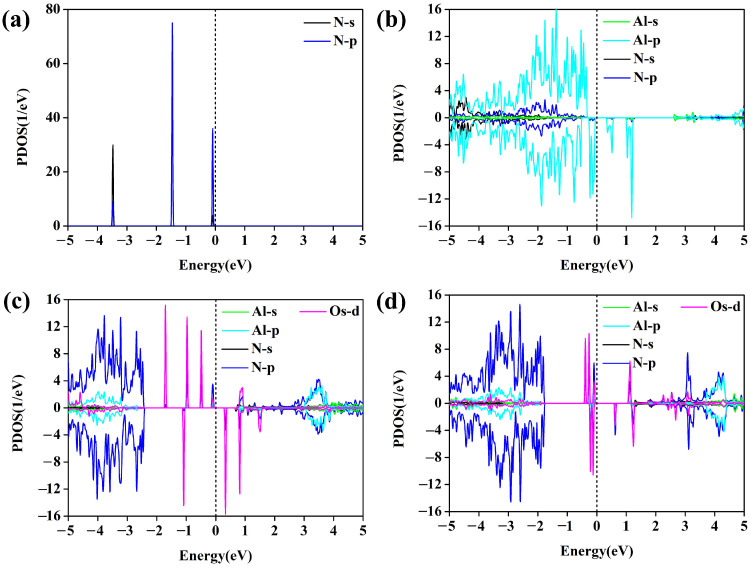
The partial density of states (PDOS) of (**a**) gaseous N_2_-s and N_2_-p, (**b**) defective AlN Al-s, Al-p, N-s and N-p, (**c**) Al-s, Al-p, N-s, N-p and Os-d of Os@AlN, and (**d**) N_2_ adsorbed on Os@AlN with end-on configuration. The Fermi level is set to zero and denoted by a black dashed line.

**Figure 7 molecules-29-05768-f007:**
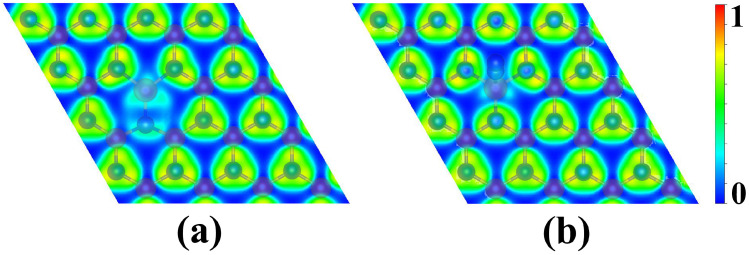
The electron localization function (ELF) maps of Os@AlN (**a**) and N_2_ adsorbed on Os@AlN with end-on configuration (**b**).

**Figure 8 molecules-29-05768-f008:**
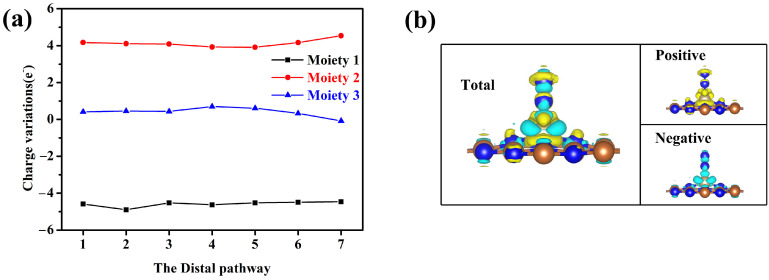
(**a**) Bader charges for three Os@AlN moieties along the distal pathway. (**b**) Charge density difference for N_2_ adsorbed on Os@AlN with end-on configuration, where the isosurface value is set to be 0.008 e Å^−3^, and the positive and negative charges are shown in yellow and cyan, respectively.

## Data Availability

The original contributions presented in this study are included in the article/Appendix A. Further inquiries can be directed to the corresponding authors.

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
