# Peer review of "Rational Design of Novel Single-Atom Catalysts of Transition-Metal-Doped 2D AlN Monolayer as Highly Effective Electrocatalysts for Nitrogen Reduction Reaction"

_molecules, 2024, doi:10.3390/molecules29235768_

Round 1

Reviewer 1 Report

Comments and Suggestions for Authors

In this manuscript, the authors reported DFT theoretical results on a wide range of single-atom catalysts consisting of transition metal atoms loaded on the two-dimensional AlN monolayer with Al monovacancy (TM@AlN) for the conversion of N2 to NH3. This work has good novelty and the results were well organized. The manuscript is considered suitable for the journal Molecules. However, to further enhance the quality and clarity of this work, the below comments need to be properly addressed

1. In the Abstract, the authors mentioned that “the limited activity and selectivity of NRR are not suitable for large-scale industrial applications”. However, the current manuscript does not address the challenge for large-scale industrial applications of NRR, therefore, this statement may not be suitable to be placed in the Abstract. A more appropriate context should be provided instead.

2. Recent works on SACs and electrocatalysis are suggested to be referenced in the Introduction (e.g., DOI: 10.1016/j.matre.2022.100144; 10.1002/inf2.12608)

3. The authors wanted to study AlN due to its many advantages such as broad band gap, high chemical and thermal stability. How is this relevant to electrocatalysis? Does it have sufficient electronic conductivity, since conductivity is a key parameter for electrocatalysis?

4. Figure 2, there are many triangles scattered in the figure. What do each of these represent? The authors label some with the Os element. How about the rest of the triangles?

Author Response

Comments 1:In this manuscript, the authors reported DFT theoretical results on a wide range of single-atom catalysts consisting of transition metal atoms loaded on the two-dimensional AlN monolayer with Al monovacancy (TM@AlN) for the conversion of N2 to NH3. This work has good novelty and the results were well organized. The manuscript is considered suitable for the journal Molecules. However, to further enhance the quality and clarity of this work, the below comments need to be properly addressed.1. In the Abstract, the authors mentioned that “the limited activity and selectivity of NRR are not suitable for large-scale industrial applications”. However, the current manuscript does not address the challenge for large-scale industrial applications of NRR, therefore, this statement may not be suitable to be placed in the Abstract. A more appropriate context should be provided instead.

Response 1: Thank you for pointing this out. We agree with this comment. Therefore, We have replaced with a more appropriate statement in the abstract.

Comments 2: Recent works on SACs and electrocatalysis are suggested to be referenced in the Introduction (e.g., DOI: 10.1016/j.matre.2022.100144; 10.1002/inf2.12608)

Response 2: Thanks a lot for the reviewer’s comment. We have cited the above literature in the introduction section.(References 24 and 25).

Comments 3: The authors wanted to study AlN due to its many advantages such as broad band gap, high chemical and thermal stability. How is this relevant to electrocatalysis? Does it have sufficient electronic conductivity, since conductivity is a key parameter for electrocatalysis?

Response 3: Thanks a lot for the reviewer’s comment. As previous literature reported (Y. Yong, Q. Zhou, X. Su, Y. Kuang, C. R. A. Catlow and X. Li, J. Mol. Liq., 2019, 289, 111153.), the electrical conductivity σ∝exp(-Eg/2kBT), where Eg is electronic band gap, kB is the Boltzmann's constant, and T is the temperature. According to the equation, the smaller band gap, the higher conductivity. After Os atom doped the AlN, the band gap reduced from the original 2.93 eV to 0.23 eV (Fig. S3, ESI). This will significantly enhances the electrical conductivity of the Os@AlN system, which been proved by similar study of TM-doped AlN systems for HER/OER/ORR electrocatalysts (Reference 38).

Comments 4: Figure 2, there are many triangles scattered in the figure. What do each of these represent? The authors label some with the Os element. How about the rest of the triangles?

Response 4: Following the reviewer’s comment, we have clearly labeled the TM@AlN system represented by each triangle in Fig. 2.

Reviewer 2 Report

Comments and Suggestions for Authors

The work is devoted to study computationally the ability of a rich series of AlN-supported transition metal atoms to act as single-atom catalysts in the reduction of N2 to ammonia. This chemical process is of major importance in the chemical industry, but the traditional catalytic conversion of N2 to ammonia (the Haber-Bosch process) suffers from major drawbacks, including the need to use massive quantities of fossil fuels and a highly unfavorable environmental impact. Developing single-atom catalysts may pave the way for the clean and sustainable large-scale ammonia production using electrochemical methods.

The authors examine the stability of 29 transition-metal atoms adsorbed on a 2D AlN support, in which an Al vacancy acts as a host site. Using a simple criterion based on protonation free energies during the N2 reduction to ammonia to screen these systems, they establish that only one case, Os@AlN, could provide a material of potential catalytic importance. They then focus on studying the electronic properties of this system, to develop a model of how an  AlN-supported could promote the activation of the highly stable N-N bond. Using Bader's analysis, they show that, when an N2 molecule coordinates to Os, electron transfer occurs from Os to the anti-bonding orbitals of the molecule, initiating its dissociation. The stability of the Os@AlN is verified using ab initio molecular dynamics simulations carried out at 350K. 

The work is focused and well carried out. The methods used, based on periodic density-functional theory calculations carried out with the VASP software, are robust and reliable. The analysis of the electronic structure provides a convincing and clear explanation concerning the N2-activation ability of Os@AlN. The paper is well written and the quality of the figures and tables (in the main text and in the supplementary material) is good. I think the paper makes an important contribution to the theoretical study of novel catalysts for N2 reduction, and could be the starting point for further work in the field. Before recommending publication, however, I would like the authors to address the minor points listed below.

1) Some of the transition metal studied are likely to be spin-polarized when adsorbed on AlN. Was spin polarization taken into account in the VASP calculations, and, if so, how was the ground spin state determined? It would be very helpful to provide in a table the spin state and transition-metal oxidation state of all the system examined.

2) Although the ab initio molecular dynamics simulations provide strong evidence that Os@AlN is stable at 350K, they cannot rule out that the system is in a meta-stable state, which could evolve to a different state (with a different conformation, for instance) in the thermodynamic limit. It would be helpful to further verify its stability by computing phonons for the optimized structure, which should not exhibit any imaginary frequencies if the system is thermodynamically stable. 

3) It is a very intriguing finding of the work that only Os seems to provide a suitable single-atom catalyst for N2 reduction. Can the authors provide some rationale, based on the electronic structure analysis that they have carried out, as to why Os seems to be so much different from all the other transition metal atoms examined? 

4) On page 5, second line from bottom of the first paragraph, replace "transform from N atom to Os" with "are transferred from the N atom to Os".

Author Response

Comments 1:Some of the transition metal studied are likely to be spin-polarized when adsorbed on AlN. Was spin polarization taken into account in the VASP calculations, and, if so, how was the ground spin state determined? It would be very helpful to provide in a table the spin state and transition-metal oxidation state of all the system examined.

Response 1: Thanks a lot for the reviewer’s comment. In this paper, we have taken into account the nonmagnetic (NM) and the ferromagnetic (FM) spin polarization during the energy and structure optimization calculations to to determine the ground state. Following the reviewer’s suggestions, we have provided the magnetic moments of FM (Mtot) of the different SACs in Table S2 of ESI and added the correlative discussions in the second paragraph of 3.1 section for the revised manuscript.

Comments 2: Although the ab initio molecular dynamics simulations provide strong evidence that Os@AlN is stable at 350K, they cannot rule out that the system is in a meta-stable state, which could evolve to a different state (with a different conformation, for instance) in the thermodynamic limit. It would be helpful to further verify its stability by computing phonons for the optimized structure, which should not exhibit any imaginary frequencies if the system is thermodynamically stable.

Response 2: Thanks a lot for the reviewer’s comment. Previous research (https://doi.org/10.1103/PhysRevB.80.155453) has conducted phonon calculations for the AlN structure. Given the large system of Os@AlN, calculating the phonon spectra is computationally intensive and time-consuming. We kindly request your understanding in this matter. Drawing from previous literature (References 10-14), the AIMD simulation provides compelling evidence that the SAC structure is thermodynamically stable.

Comments 3: It is a very intriguing finding of the work that only Os seems to provide a suitable single-atom catalyst for N2 reduction. Can the authors provide some rationale, based on the electronic structure analysis that they have carried out, as to why Os seems to be so much different from all the other transition metal atoms examined?

Response 3: Thanks a lot for the reviewer’s comment. First, after screening by the criterion that both the values of the free energy change of the first protonation (ΔG(*N2–*NNH)) or the last protonation steps (ΔG(*NH2–*NH3)) should be less than 0.49 eV for highly active NRR catalysts. Only Os@AlN meets this criterion. Second, the calculated binding energy of Os@AlN is quite negative (-11.623 eV), indicating excellent thermodynamic stability. This can be confirmed by the ab initio molecular dynamics simulations at 350 K. Third, as shown in Figs. 7 and 8, when N2 is adsorbed on Os@AlN in an end-on adsorption configuration, significant electron transfer occurs between Os@AlN and N2, forming the ‘σ-donation and π-back-donation’ electron transfer mechanism. This leads to orbital hybridization in N2-Os@AlN (Fig. 6), which makes the subsequent hydrogenation steps easier. Above all, the uniqueness of Os@AlN merits further in-depth study.

Comments 4: On page 5, second line from bottom of the first paragraph, replace "transform from N atom to Os" with "are transferred from the N atom to Os".

Response 4: Thank you for pointing this out. Following the reviewer’s suggestions, we have substituted the corresponding sentences.